# Titanium Dioxide Nanoparticles Enhance Leakiness and Drug Permeability in Primary Human Hepatic Sinusoidal Endothelial Cells

**DOI:** 10.3390/ijms20010035

**Published:** 2018-12-21

**Authors:** Jie Kai Tee, Li Yang Ng, Hannah Yun Koh, David Tai Leong, Han Kiat Ho

**Affiliations:** 1Department of Pharmacy, Faculty of Science, National University of Singapore, 18 Science Drive 4, Singapore 117543, Singapore; jiekai.tee@u.nus.edu (J.K.T.); ngliyang.95@gmail.com (L.Y.N.); hannahkyy@gmail.com (H.Y.K.); 2NUS Graduate School for Integrative Sciences & Engineering, Centre for Life Sciences, 28 Medical Drive, Singapore 117456, Singapore; cheltwd@nus.edu.sg; 3Department of Chemical and Biomolecular Engineering, National University of Singapore, 4 Engineering Drive 4, Singapore 117585, Singapore

**Keywords:** cell–material interaction, titanium dioxide nanoparticles, liver sinusoidal endothelial cells, endothelial permeability, oxidative stress, Akt pathway, cell attachment, liver fibrosis

## Abstract

Liver sinusoidal endothelial cells (LSECs) represent the permeable interface that segregates the blood compartment from the hepatic cells, regulating hepatic vascular tone and portal pressure amidst changes in the blood flow. In the presence of pathological conditions, phenotypic changes in LSECs contribute to the progression of chronic liver diseases, including the loss of endothelial permeability. Therefore, modulating LSECs offers a possible way to restore sinusoidal permeability and thereby improve hepatic recovery. Herein, we showed that titanium dioxide nanoparticles (TiO_2_ NPs) could induce transient leakiness in primary human hepatic sinusoidal endothelial cells (HHSECs). Interestingly, HHSECs exposed to these NPs exhibited reduced protein kinase B (Akt) phosphorylation, an important protein kinase which regulates cell attachment. Using a 3D co-culture system, we demonstrated that TiO_2_ NPs diminished the attachment of HHSECs onto normal human hepatic cell LO2. To further illustrate the significance of leakiness in liver sinusoids, we showed that NP-induced leakiness promoted Sunitinib transport across the HHSEC layer, resulting in increased drug uptake and efficacy. Hence, TiO_2_ NPs have the potential to modulate endothelial permeability within the specialized sinusoidal endothelium, especially during events of fibrosis and occlusion. This study highlighted the possible use of inorganic NPs as a novel strategy to promote drug delivery targeting the diseased liver.

## 1. Introduction

The liver is a vital organ involved in homeostasis through the metabolism of biomolecules and xenobiotics, regulation of blood glucose, bile production and excretion of bilirubin. With high exposure to circulating endotoxins and antigens from the gut, the liver also plays a central role in the immunological response to remove toxic agents from the bloodstream [1]. These functions are supported by the presence of discontinuous sinusoids formed by fenestrated endothelium with intercellular gaps and fragmented basement membrane to facilitate the bidirectional transport of cells and molecules across the sinusoidal barrier [2]. Lining the endothelium, liver sinusoidal endothelial cells (LSECs) are highly specialised cells that constitute the permeable barrier between the blood and liver parenchyma [3]. Under physiological conditions, they help to maintain portal pressure by regulating hepatic vascular tone and contribute to hepatic stellate cell (HSC) quiescence, thereby inhibiting fibrogenesis and intrahepatic vasoconstriction. However, pathological factors such as viral infections, alcoholism and schistosomiasis initiate a highly conserved wound healing response known as liver fibrosis, resulting in the activation of HSCs and excessive secretion of extracellular matrix (ECM) in the liver [4,5]. As the LSECs undergo capillarisation during liver fibrosis, profound morphological and functional changes such as the loss of fenestrae and the development of a basement membrane occur [6,7]. Coupled with the accumulation of ECM in the perisinusoidal space and loss of endothelial leakiness, intrahepatic resistance increases and ultimately leads to the collapse of the sinusoids. As a result, portal hypertension and hepatic dysfunction ensue as the major consequences of liver cirrhosis [8].

Early intervention of liver fibrosis is the key to preventing the onset of chronic liver disease and to promoting liver regeneration. The underlying mechanisms include the reversion of fibrogenic cells, degradation of ECM and a switch in the inflammatory environment [9,10]. These processes promote the restoration of liver sinusoids, thereby alleviating the fibrotic condition and contributing to hepatic recovery. Henceforth, studies have developed novel therapeutic strategies to ameliorate blood circulation in the liver and reduce complications associated with liver fibrosis [11]. Growth factors such as vascular endothelial growth factor (VEGF) and hepatocyte growth factor (HGF) have been demonstrated to alleviate inflammatory conditions and improve blood vessel formation while promoting hepatic tissue repair [12,13]. Drugs such as statins and obeticholic acid have been shown to modulate liver microvascular dysfunction by increasing nitric oxide (NO) bioavailability to promote vasodilation and reduce hepatic vascular tone [14]. However, many of these strategies failed to retain an effective local concentration due to their rapid diffusion to extracellular fluids and short half-lives.

The use of inorganic nanoparticles (NPs) offers an alternative strategy to overcome the limitations of biological factors and drug molecules [15] while modulating the diseased endothelium during fibrotic progression. Previous studies have shown that titanium dioxide NPs (TiO_2_ NPs) could induce endothelial leakiness through the disruption of vascular endothelial-cadherin (VE-cadherin) and the remodelling of actin cytoskeleton [16,17]. Nanodiamonds (NDs) [18] and iron oxide NPs [19] could increase endothelial permeability through the production of reactive oxygen species (ROS) and the remodelling of microtubules. High-density silica NPs (SiO_2_ NPs) were shown to perturb endothelial cell–cell contacts by exerting a mechanic force onto the pre-tensed VE-cadherin that holds the endothelial cells together. Gold NPs (Au NPs) were shown to induce variable endothelial leakiness depending on the particle size [20]. Aside from the ability to induce endothelial leakiness, inorganic NPs have also been regarded as effective inhibitors for cellular hepatic fibrosis [21]. This further exemplified the therapeutic potential of inorganic NPs to treat liver fibrosis. While these studies highlighted the possibility of using inorganic NPs to remodel the endothelium and promote vessel leakiness, there are currently no reports suggesting the use of inorganic NPs to restore the endothelial leakiness in the fibrotic liver.

Herein, we showed that TiO_2_ NPs induced endothelial leakiness in primary human hepatic sinusoidal endothelial cells (HHSECs). TiO_2_ NPs were rapidly internalized into HHSECs within 30 min exposure and did not affect cell viability up to 72 h. Interestingly, HHSECs exposed to TiO_2_ NPs exhibited a reduction in phosphorylated Akt, a molecular process involved in cellular detachment. Using a 3D co-culture model of normal human hepatic cells (LO2) and HHSECs, we showed that TiO_2_ NPs promoted the breakaway of cells from the core culture. This phenomenon supported the increased transport of Sunitinib across a fibrotic HHSEC layer, thereby enhancing drug permeability and drug response. Therefore, this research highlights a potential use of TiO_2_ NPs to manipulate sinusoidal leakiness for therapeutic applications during fibrotic progression.

## 2. Results

### 2.1. TiO_2_ NPs Induced Endothelial Leakiness in the Sinusoidal Barrier

We have observed that inorganic NPs are capable of inducing leakiness in various endothelial cell models [20], but this effect may not be generalizable across all endothelial linings. Given that vascular barriers formed by sinusoidal endothelial cells are known to be permeable and discontinuous, we asked whether TiO_2_ NPs could further enhance the leakiness in the HHSEC-formed endothelial layer. We first measured the hydrodynamic size and zeta potential of spherical TiO_2_ NPs in media, and they were shown to be relatively stable up to 90 min (Figure 1A,B). This is important in order to ascertain that effects on endothelial leakiness were not attributed to progressive changes in the NP size and surface zeta potential over time. Compared to the endothelial layer established by human microvascular endothelial cells (HMVECs), we found that HHSECs exhibited a 1.5-fold increase in endothelial leakiness (Figure 2A). The enhanced leakiness was not significantly reduced with the additional coating of fibronectin, which was used for all subsequent experiments to enhance the attachment of endothelial cells. HHSECs treated with TiO_2_ NPs for 30 min at concentrations of 100 µM and 500 µM were shown to exhibit a significantly higher leakiness compared to the non-treated control (Figure 2B). We first considered how TiO_2_ NPs could perturb the characteristic biomarkers of HHSECs (Appendix A), thereby resulting in a change in phenotype that led to endothelial leakiness. Hence, HHSECs were treated with TiO_2_ NPs up to 72 h, and a combination of biomarkers previously used to identify HHSECs [22,23] from other endothelial cells, namely vascular endothelial growth factor receptors (VEGFR-1, VEGFR-2, VEGFR-3), VE-cadherin, endothelial tyrosine kinase (Tie-2), platelet endothelial cell adhesion molecule (CD31) and Stabilin-1 (Stab-1), were assessed (Figure 2C). The protein levels of these tested biomarkers did not vary significantly at earlier time points of 24 and 48 h. However, there were observable changes to the expression of VEGFR-1, VEGFR-2 and VE-cadherin only at 72 h. This prompted us to further investigate the biological effects of TiO_2_ NPs on HHSECs.

### 2.2. Endothelial Leakiness Was Not Due to a Decrease in Cell Viability

Endothelial leakiness could possibly be attributed to the toxic effects of inorganic NPs in biological systems. To rule out this possibility, we treated HHSECs with various concentrations (50–1000 µM) of TiO_2_ NPs up to 72 h and found no significant reduction in cell viability for all three timepoints (Figure 2D). Notably, we observed a concentration-dependent decrease in cell viability at 72 h, suggesting that TiO_2_ NPs may reduce the proliferation of HHSECs with prolonged exposure, albeit not statistically significantly. Cell imaging revealed that TiO_2_ NPs caused the shrinkage and detachment of HHSECs from the surface, thereby resulting in the formation of large gaps between the cells (Figure 2E). This effect was more obvious in HHSECs treated with a higher concentration (500 µM) of TiO_2_ NPs. We noticed that HHSECs were not able to form a uniform monolayer even after incubation for 72 h. Their tendency to lose contacts with neighbouring cells over time suggested the formation of leaky endothelium when culturing for longer periods of time.

### 2.3. Internalised TiO_2_ NPs Did Not Significantly Promote Oxidative Stress

With the high capacity of HHSECs to endocytose foreign particles [3], we investigated whether TiO_2_ NPs could similarly be internalised into the cells. Using fluorescein isothiocyanate (FITC)-conjugated TiO_2_ NPs for fluorescence visualisation, we observed localisation of TiO_2_ NPs within the cell after 30 min of treatment (Figure 3A). TiO_2_ NPs were found to co-localise with lysosomes even with the co-treatment of endocytosis inhibitors monodansylcadaverine (MDC) and methyl-β-cyclodextrin (MβCD) (Appendix A). The internalised TiO_2_ NPs at 30 min did not reorganise the actin fibres; an effect that was commonly seen in other NP-induced endothelial leakiness [24,25]. However, the actin structures appeared to be more disorganised when HHSECs were treated for 3 h, particularly at regions where TiO_2_ NPs were localised. With the increase in endothelial permeability coupled with the remodelling of the actin fibres, we further questioned the underlying mechanisms for this effect. Intuitively, the observed morphological changes could arise as a result of physical stress or biochemical response. To determine the presence of a biochemical trigger, we explored evidence for oxidative stress as an early event. We first measured the intracellular ROS levels using H_2_DCF-DA oxidative stress indicator and observed a subtle increase in ROS production with increasing concentrations of TiO_2_ NPs (Figure 3B). However, this marginal change to the oxidative level may not account for the observed changes in the morphology of HHSECs when exposed to TiO_2_ NPs. To support this notion, we further assess the expression levels of inflammatory markers such as nuclear factor kappa-light-chain-enhancer of activated B cells (NFκB), cyclooxygenases (COX-1, COX-2) and oxidative stress markers such as nuclear factor (erythroid-derived 2)-like 2 (NRF-2) and heme oxygenase-1 (HO-1) at 3 h of exposure to TiO_2_ NPs. The protein levels of all markers did not change in response to TiO_2_ NP treatment (Appendix A). Instead, we noticed an increase in expression of COX-2 and HO-1 in the presence of endocytosis inhibitors, suggesting that oxidative balance was maintained during exposure to TiO_2_ NPs.

### 2.4. TiO_2_ NPs Weakened the Attachment of Endothelial Cells

While we observed the cellular shrinkage of HHSECs upon exposure to TiO_2_ NPs (Figure 2E), we postulated that there are certain cell signalling changes within the cell that could mediate this effect. Protein kinase B (Akt) signalling is one of the major pathways involved in endothelial cell attachment onto the substratum [26,27]. Therefore, a decrease in Akt phosphorylation would signal a reduction in cellular attachment. We treated HHSECs with TiO_2_ NPs at varying timepoints and assessed the phosphorylation of Akt. Cells treated with TiO_2_ NPs exhibited a reduction in phosphorylated Akt, particularly at 3 and 6 h of NP exposure (Figure 4A). NFκB levels were also assessed to verify the absence of inflammation that could account for the induction of endothelial leakiness at short exposures. Since Akt phosphorylation started to reduce between 15 min and 3 h of TiO_2_ NP exposure, we captured a more detailed time-course analysis at 15 min intervals up to 60 min (Figure 4B). The pre-treatment of cells with Akt inhibitor (LY294002) was used as a positive control to inhibit Akt phosphorylation. Indeed, we observed a time-dependent decrease in Akt phosphorylation upon TiO_2_ NP exposure. This finding coincides with the notion that a reduction in phosphorylated Akt would signal the decrease in cellular attachment. Furthermore, intracellular calcium release downstream of Akt, coupled with the induction of oxidative stress, is often thought to mediate endothelial leakiness as this process precedes morphological changes in endothelial cell shape [18,28]. Therefore, we continued to measure intracellular calcium ions (Ca^2+^) in the cell lysates. Unexpectedly, the exposure of HHSECs to two different concentrations of TiO_2_ NPs did not promote any observable increase in intracellular Ca^2+^ (Figure 4C). This suggested that the endothelial leakiness observed was due to other effects mediated by a reduction in Akt phosphorylation.

### 2.5. TiO_2_ NPs Enhanced the Dissociation of Cell–Cell Contact in a 3D Spheroid Co-Culture Model

Inspired by 3D spheroid co-culture systems of hepatocytes (LO2 cells) and liver endothelial cells (HHSECs) as a better representation of the in vivo liver construct [29,30,31,32,33], we further tested our hypothesis by establishing a co-culture model with fluorescence-labelled hepatic LO2 cells and HHSECs. After the formation of the 3D spheroid consisting of LO2 cells as the core and HHSECs as the shell, we further incubated the co-cultured model with TiO_2_ NPs for 24 h. Interestingly, we observed a detachment of cells, particularly the HHSECs (green cells) from the LO2 core (red cells) in both 100 µM and 500 µM TiO_2_ NP-treated groups (Figure 5A). The disruption of the 3D-spheroid surface was more pronounced at a higher concentration (500 µM) of TiO_2_ NPs. As the HHSECs appeared to be losing HHSEC–LO2 cell–cell contacts and migrating away from the LO2 core, the core increases its own internal cell–cell contact such that the spheroid core shrinks, resulting in a higher-intensity fluorescent red image. Using Imaris software to reconstruct a cross-sectional image of the 3D spheroid, we found that HHSECs started to break away from the surfaces of the spheroid as well as an observable tightening of the LO2 cells within the core in the TiO_2_ NP-treated groups (Figure 5B). Henceforth, these results suggested that TiO_2_ NPs weakened the attachment of HHSECs from the endothelial layer, resulting in leakiness.

### 2.6. Endothelial Leakiness Promotes Anti-Fibrotic Therapy

Liver fibrosis is a condition worsened by the loss of endothelial leakiness and excessive deposition of ECM lining the sinusoids. Therefore, strategies to weaken the attachment of sinusoidal endothelial cells stranded in the ECM layer during liver fibrosis could provide an alternative way to favour anti-fibrotic therapy through the increase in endothelial permeability. Henceforth, the clinical benefits of such enhanced endothelial leakiness could include improving the drug delivery across the sinusoidal endothelium during treatment of liver fibrosis, especially when the ECM-rich barrier impairs drug clearance by limiting the amounts of therapeutic drugs reaching the liver parenchyma. To support this notion, we first optimised the seeding of HHSECs on transwell inserts coated with collagen and fibronectin to simulate the fibrotic condition of the endothelium (Appendix A). The resulting endothelium exhibited a reduction in endothelial leakiness as compared to the HHSEC layer which was not pretreated with the substrate (no fibronectin). Using this fibrotic model, we showed that 100 µM TiO_2_ NPs could increase endothelial permeability within 30 min (Appendix A). This NP-induced leakiness required the presence of HHSECs as we showed that there was no increase in leakiness in the insert with only the ECM coating, which further suggested that the ECM layer barrier was compromised by the endothelial cells. In order to demonstrate the application of NP-induced leakiness in the context of drug delivery, we first treated the HHSEC layer with TiO_2_ NPs for 30 min to induce endothelial leakiness before assessing the transport of Sunitinib across the leaky endothelium. After incubation with Sunitinib for another 30 min, we detected a significantly higher concentration of Sunitinib from a range of 1 µM to 1.24 µM across the TiO_2_ NP-treated endothelium, suggesting that an increased amount of Sunitinib have diffused across the endothelial barrier (Figure 6A). Given that the increase was approximately 1.24-fold, we further demonstrated the efficacy of our strategy by using the relative concentrations of Sunitinib to treat transforming growth factor-beta 1 (TGF-β1)-activated hepatic stellate cells LX2, the key fibrotic cell that drives liver fibrosis [4]. Interestingly, there was an observable reduction in TGF-β1-induced α-smooth muscle actin (αSMA) upon exposure from 1 to 2 µM of Sunitinib (Figure 6B), a key biomarker of stellate cell activation and subsequent fibrosis. The therapeutic effects of such increase in Sunitinib concentration was further highlighted in our 3-(4,5-dimethylthiazol-2-yl)-2,5-diphenyltetrazolium bromide (MTT) assay, in which we showed a significant reduction of LX2 viability beyond 1 µM of Sunitinib (Figure 6C). Therefore, such an increase in Sunitinib concentration caused by a subtle increase in endothelial leakiness could be translated to a significant enhancement of drug efficacy.

## 3. Discussion

The development of a fibrotic liver encompasses a spectrum of biological changes in the microenvironment which converge on a pivotal event of diminished blood flow through the sinusoidal vessels. This phenomenon not only compromises intrahepatic functions (e.g., xenobiotic clearance), but can also trigger extrahepatic problems (e.g., portal hypertension and edema). Strategies to modulate liver microvascular dysfunction and regulate vascular tone could thus help to alleviate the fibrotic condition. In this study, we first demonstrated that TiO_2_ NPs could enhance endothelial leakiness in HHSEC-formed endothelium within a short time frame. Secondly, endothelial leakiness was not accompanied with the induction of oxidative stress, a phenomenon normally observed with the use of inorganic NPs in biological systems [34]. Thirdly, HHSECs treated with TiO_2_ NPs exhibited a profound reduction in Akt phosphorylation, which potentially mediates the loss of cellular attachment from the substratum, resulting in further leakiness. With these key points in mind, we anticipate that such biochemical perturbations could be further exploited to promote the delivery of anti-fibrotic drugs such as Sunitinib across the leaky sinusoidal barrier.

Although previous studies have shown that NP-induced endothelial leakiness was attributed to the loss of VE-cadherin at the cell–cell contacts [16,35], we observed a reduction in VE-cadherin expression only at 72 h (Figure 2C), independent from the enhanced leakiness observed within 30 min (Figure 2B). VE-cadherin is an adherens junction protein that plays an important role in maintaining the integrity of endothelial cell junctions [36]. Despite being the major cadherin expressed by HHSECs, the relatively low expression of VE-cadherin could be due to the lack of classical adherens junctions in the sinusoidal endothelium [22]. This accounts for the leaky phenotype observed in HHSECs as well as their limited involvement during NP-induced endothelial leakiness. In order to observe an even leakier phenotype in the HHSECs layer, TiO_2_ NPs would have significantly altered the morphology of the endothelial cells. As such, we postulated that TiO_2_ NPs could have weakened cellular attachment from the substratum. Wang et al. showed that negatively charged Au NPs induced greater endothelial leakiness than their positive counterparts due to a series of repulsive–sedimentary interactions between the inorganic NPs and glycocalyx found on the cell surface, both of which are negatively charged [37]. Similarly, Voijt et al. demonstrated that polyanionic lipid NPs exert electrostatic repulsion to promote hydrophobic interaction with the lipid raft structure of caveolae, which are small invaginations highly expressed on the endothelial cell membrane [38]. Hence, endothelial leakiness could be induced especially if the NP is negatively charged; in this case, 100 µM TiO_2_ NPs exhibited a zeta potential of -10 mV in media (Figure 1A,B). Therefore, the TiO_2_ NP-induced endothelial leakiness observed in our current study could well be attributed to strong NP–cell interactions which resulted in the separation of endothelial cells away from each other.

The application of inorganic NPs on the liver sinusoids during fibrosis poses as a double-edged sword. On one hand, NPs could help to modulate endothelial permeability and restore vascular tone. On the other hand, the excessive use of NPs could lead to undesirable biological stress on the endothelial cells. Henceforth, inducing endothelial leakiness without exerting significant cellular toxicity and other adverse consequences is an important therapeutic consideration to promote the clinical use of inorganic NPs. Our Western blot analyses (Figure 2C) revealed that these NPs did not induce endothelial dysfunction as several of these endothelial biomarkers were not significantly perturbed. Furthermore, our MTT cell viability assays (Figure 2D) showed that TiO_2_ NPs up to 1000 µM did not induce significant reduction in the proliferation of HHSECs. Instead, there was observable cell proliferation at all NP concentrations over the treatment time up to 72 h, suggesting that TiO_2_ NPs did not affect the ability of the cells to proliferate. In addition, treatment of TiO_2_ NPs did not induce significant ROS production (Figure 3B) or exert any observable changes to the protein biomarkers of oxidative stress (Appendix A). Therefore, we rule out the possibility that TiO_2_ NPs could affect cell survival within these time points. In other words, the induced endothelial leakiness was not attributed to the reduction in cell viability or generation of oxidative stress. This is in stark contrast to other established studies linking oxidative stress and endothelial permeability, in which the degree of leakiness is attributed to the ROS production profile. Liu et al. has showed that SiO_2_ NPs disrupted the blood–brain barrier through the loss of tight junctions and cytoskeleton arrangement, accompanied by an increase in inflammatory response and presence of oxidative stress [39]. Nevertheless, such biological stress can be regulated through the surface functionalisation of NPs to induce a tunable endothelial permeability favourable for drug delivery strategies [18]. As such, we speculated that a well calibrated use of TiO_2_ NPs could achieve a favourable stress response which then leads to the desired endothelial permeability.

Our results demonstrated that TiO_2_ NPs can be used at a lower concentration (100 µM) to induce a similar leakiness effect compared to higher concentrations. This ensures that the targeted HHSECs could retain their physiological roles in mediating an important protective function as an endothelial barrier of the liver sinusoids while minimizing the bioaccumulation of NPs [40]. A study has demonstrated that 40 μg/mL of 20 nm Au NPs caused a 50% increase leakiness in human umbilical vein endothelial cells (HUVECs) and a 40% decay in cell–ECM adhesive forces without affecting cell viability significantly [24]. The observed endothelial leakiness was induced through unbalanced intracellular tensions and paracellular forces, coupled with actin rearrangement mediated by the Rho-associated protein kinase (ROCK)-dependent pathway. This study coincides with our endothelial permeability (Figure 2B) and cell attachment experiments (Figure 5A,B) that NPs could weaken cell–cell and cell–ECM contacts to promote endothelial leakiness. Our immunofluorescence results (Figure 3A) also revealed the disruption of the actin cytoskeleton, especially at sites where the TiO_2_ NPs were internalised, suggesting the close interaction of actin fibres and the NPs in contributing to the phenotypic changes. Furthermore, we hypothesized that TiO_2_ NPs could induce leakiness by reducing the activation of Akt (Figure 4A,B), a major pathway involved in cell survival and attachment. Indeed, inorganic NPs such as SiO_2_ NPs have been previously shown to disrupt endothelial cell function and promote vascular permeability through the inhibition of the phosphoinositide 3-kinase (PI3K)/Akt signalling pathway [41]. Cell attachment studies on endothelial tyrosine kinase Tie2-mediated anchoring have also reported the significance of the Akt pathway to provide a stabilizing signal in the endothelial cells [26], as well as being preferentially activated by dimerized Tie2 in the presence of cell–cell contacts [27]. Furthermore, other studies have reported Akt-dependent mechanisms involving the downregulation of eNOS activity which could direct the loss of cellular attachment from the substratum [42,43]. Taken together, these studies have specifically pinpointed the involvement of Akt in cellular detachment. Therefore, our findings are consistent with these observations, where a mechano-transduction process may drive TiO_2_ NP-induced endothelial leakiness towards a structural and a functional change.

Endothelial leakiness has been phenotypically attributed to the loss of cell–cell contacts due to the disruption of adherens junction proteins [16], cytoskeleton rearrangement resulting in cell retraction [18], or the weakening of cell adhesion from the substratum [24]. Although 2D transwell assay is regarded as a gold standard technique to assess endothelial barrier function [44], this assay may not capture the key finding of cell detachment that addressed the leaky phenomenon. Henceforth, endothelial leakiness was further shown in the 3D co-culture model comprising hepatic LO2 cells and HHSECs in which the exposure to TiO_2_ NPs caused the disruption of the core–shell model and the dispersion of HHSECs away from the structure. (Figure 5). The co-culturing of hepatocytes and endothelial cells has been previously reported to sustain the long-term viability of cells while mimicking the liver sinusoid conditions. Kang et al. demonstrated that primary rat hepatocytes and LSECs can be cultured together with cells exhibiting normal morphology up to 39 days [30]. Another study has also showed the use of tricultures comprising of hepatocytes, fibroblast and endothelial cells to elucidate reciprocal cellular interactions between the different cell types [45]. From our study, we demonstrated that TiO_2_ NP-induced endothelial leakiness is not limited to monolayer cultures as the results showed substantial detachment of cells from the LO2/HHSEC co-culture model (Figure 5). This finding confirmed the decrease in cell-cell interactions, thereby disrupting the intricate 3D conformation.

In order to widen the applicability of enhanced leakiness caused by TiO_2_ NPs, we assessed whether such leakiness could promote the transport of therapeutic agents across the HHSEC layer. This phenomenon is particularly important in diseases featuring a reduction in endothelial permeability, such as liver fibrosis. During fibrotic progression, multiple cellular changes occur in the liver microenvironment, including a self-perpetuating cycle between the capillarized LSECs and activated HSCs which stimulate each other [46]. As a result, the loss of the basement membrane and excessive secretion of ECM resulted in a drastic reduction in endothelial leakiness within the liver sinusoids. To prove that the TiO_2_ NP-induced endothelial leakiness provides a window of opportunity for therapeutic drugs to enter, we first constructed a monolayer of HHSECs seeded onto collagen and fibronectin-coated transwell. A significant reduction of endothelial leakiness was observed in the collagen/fibronectin-coated transwell, which mimicked the fibrotic condition of the liver sinusoid (Appendix A). The induction of endothelial leakiness by TiO_2_ NPs was observed to be cell-dependent (Appendix A), thus showing that cellular movement is required to provide the necessary force to dislodge the cells from attachment to the ECM or from neighbouring cells. Using Sunitinib as the drug model, we further demonstrated that NP-induced endothelial leakiness could promote drug transport across the endothelium to enhance the efficacy of anti-fibrotic therapeutics (Figure 6A). Sunitinib is a well-known multitarget tyrosine kinase inhibitor commonly investigated for its anti-fibrotic effects [47]. Enhancing endothelial leakiness may help to overcome the therapeutic limitations imposed by liver fibrosis where the uptake and accumulation of drugs within the cells is compromised [48]. In the presence of TiO_2_ NP treatment, we demonstrated a 1.24-fold increase in Sunitinib concentration at the bottom of the transwell chamber (Figure 6A). Clearly, there was an observable decrease in both the fibrotic markers (Figure 6B) and cellular viability (Figure 6C) of activated LX2 upon exposure to Sunitinib, even at low doses of 1 to 2 µM, suggesting that a minute increase of endothelial leakiness may still result in a significant enhancement of drug efficacy. Henceforth, these findings reveal a possible use of NP-induced endothelial leakiness as a relatively safe and exploitable strategy to improve drug delivery across an otherwise impenetrable vascular barrier in diseased conditions.

## 4. Materials and Methods

### 4.1. Cell Culture and Reagents

Human hepatic sinusoidal endothelial cells (HHSECs) (ScienCell Research Laboratories, Carlsbad, CA, USA) were cultured in endothelial cell medium (ScienCell Research Laboratories, CA, USA) supplemented with 5% fetal bovine serum (FBS) according to the manufacturer’s instructions. Human hepatic cells, LO2, were received as a kind gift from A/Prof. Victor Yu (National University of Singapore, Department of Pharmacy, Singapore) and cultured in high glucose Dulbecco’s modified Eagle’s medium (DMEM) (Sigma-Aldrich, St. Louis, MO, USA) supplemented with 10% FBS. Human hepatic stellate cells, LX2, were received as a kind gift from Prof. Scott Friedman (Mount Sinai Hospital, New York, NY, USA) and cultured in low-serum DMEM supplemented with 1% FBS. All cells were incubated in a humidified atmosphere at 37 °C with 5% CO_2_. LX2 cells were activated with 2 ng/mL transforming growth factor beta 1 (TGF-β1) (Merck Millipore, Burlington, MA, USA) for 24 h. The following inhibitors were used: monodansylcadaverine (MDC), methyl-β-cyclodextrin (MβCD) (Sigma-Aldrich, St. Louis, MO, USA) and LY294004 (Cell Signaling Technology, Danvers, MA, USA). The following antibodies were used: anti-Flt-1/VEGFR-1, anti-Flk-1/VEGFR-2, anti-Flt-4/VEGFR-3, anti-stabilin-1, anti-COX-1 and anti-COX-2 (Santa Cruz Biotechnology, Dallas, TX, USA); anti-VE-cadherin, anti-Tie-2 (Western blot), anti-CD31, anti-NFκB, anti-HO-1, anti-phospho-Akt, anti-Akt, anti-GAPDH, secondary anti-mouse and anti-rabbit (Cell Signaling Technology, MA, USA); anti-Tie-2 (Imaging) (Sino Biological, Beijing, China); anti-NRF-2 (MBL international, Woburn, MA, USA); anti-αSMA (Agilent Dako, Santa Clara, CA, USA).

### 4.2. Preparation of TiO_2_ NPs

Titanium dioxide nanoparticles (TiO_2_ NPs) were first prepared by dispersing 5 mg of p25 TiO_2_ nanopowder (MR: 79.87) with primary size of 21 nm (Sigma-Aldrich, St. Louis, MO, USA) in 1 mL of water to give 62.5 mM concentration and sonicated using Sonic Dismembrator model 100 (Thermo Fisher Scientific, Waltham, MA, USA) for 45 s. Subsequently, the suspension was further diluted with endothelial cell media (ScienCell Research Laboratories, CA, USA) into either 100 µM or 500 µM and sonicated for another 45 s. Hydrodynamic size and surface zeta potential of the NPs were then measured using Zetasizer Nano ZS90 size analyzer (Malvern Panalytical, Malvern, England). FITC was covalently tagged onto TiO_2_ NPs according to the previous study [16].

### 4.3. Transwell Permeability Assay

Endothelial cells were seeded at a density of thirty thousand cells/cm^2^ onto 24-well 6.5 mm transwell insert, polyester filter with 0.4 µm pores (Corning, Corning, NY, USA) for 3 days to form a confluent layer. Thereafter, TiO_2_ NPs were mixed together with 1 mg/mL fluorescein isothiocyanate (FITC)-dextran (MW: 40,000; Sigma-Aldrich, St. Louis, MO, USA) in media and added to the cells for 30 min. Subsequently, 100 µL of media was collected from the lower compartment of the transwell and placed into a 96-well black plate. Fluorescence signal was then quantified using Hidex Sense microplate reader (Hidex, Turku, Finland) at Ex490/Em520 nm and data was normalised with the untreated control.

### 4.4. Western Blot

Cells were harvested and lysed with radioimmunoprecipitation assay (RIPA) buffer containing 0.1% sodium dodecyl sulfate (SDS) w/v, 0.5% Sodium deoxycholate w/v and 1% NP-40 v/v. Thereafter, protein lysates were loaded into 10% SDS-PAGE polyacrylamide gel (Bio-Rad Laboratories, Hercules, CA, USA) and run at 130 V for 2 h. Subsequently, a “sandwich” method was used to transfer the proteins onto polyvinylidene difluoride (PVDF) membrane (Thermo Fisher Scientific, Waltham, MA, USA) at 100V for 2 h at 4 °C. The membrane was then block with 5% bovine serum albumin w/v in Tris-buffered saline (1st Base, Singapore) containing 0.1% tween v/v (TBST) for 1 h. After blocking, the membrane was incubated with the respective primary antibody (1:1000) diluted with 2% BSA w/v in TBST overnight. The membrane was washed thrice with TBST and incubated with secondary antibody (1:10,000) for 1 h. After washing the membrane again thrice with TBST, Western Lightning Plus-ECL reagent (Perkin Elmer, Waltham, MA, USA) was added and the membrane was exposed with G:Box Gel imaging system (Syngene, Bangalore, India).

### 4.5. Cell Viability Assay

The 3-(4,5-dimethylthiazol-2-yl)-2,5-diphenyltetrazolium bromide (MTT) assay (Duchefa Biochemie, Haarlem, Netherlands) was performed to measure cell viability. Five thousand cells/well were first seeded in a 96-well plate overnight before treatment with the respective compounds at different concentrations for 24, 48 and 72 h. After treatment, the media was removed and replaced with fresh media containing 0.5 mg mL^−1^ of MTT dye for another 3 h at 37 °C. The dye was then removed and 200 µL of dimethyl sulfoxide (DMSO) (Fisher Scientific, Hampton, NH, USA) was added to dissolve the formazan salt. Absorbance was measured using Hidex sense microplate reader (Hidex, Turku, Finland) and data was normalised with the untreated control.

### 4.6. Immunofluorescence Imaging

Five thousand cells/well of HHSECs seeded onto 8-well chamber slides (Thermo Fisher Scientific, Waltham, MA, USA) were fixed with 4% paraformaldehyde w/v (Sigma-Aldrich, St. Louis, MO, USA) and permeabilized with 0.2% Triton X-100 v/v in PBS (Sigma-Aldrich, St. Louis, MO, USA) for 15 min each. Subsequently, the cells were blocked with 2% BSA w/v in PBS on ice for 1 h. After blocking, respective primary antibody (1:200) diluted in 0.2% w/v BSA and 0.1% v/v Triton X-100 in PBS was added and incubated overnight at 4 °C. Cells were washed twice with PBS and incubated with CF568 phalloidin (Biotium, Fremont, CA, USA), Hoechst 33342 dye (Sigma-Aldrich, St. Louis, MO, USA) and/or Alexa 488 chicken anti-rabbit/mouse (1:400) for 1 h at room temperature. After washing twice with PBS, 10 µL of Fluoromount^TM^ aqueous mounting medium (Sigma-Aldrich, St. Louis, MO, USA) was added onto each sample and fluorescence images were captured with Olympus Fluoview FV1000 confocal microscope (Olympus, Tokyo, Japan).

### 4.7. Oxidative Stress Assay

General oxidative stress indicator (CM-H_2_DCFDA) assay (Thermo Fisher Scientific, Waltham, MA, USA) was performed to assess the production of ROS in HHSECs upon exposure to TiO_2_ NPs. ten thousand cells/well were seeded onto 96-well black, clear bottom plate for 2 days. After treatment with TiO_2_ NPs, the cells were washed with PBS and incubated with 10 µM CM-H_2_DCFDA reagent and Hoechst 33342 dye (Sigma-Aldrich, St. Louis, MO, USA) for 30 min. The reagents were removed and cells were wash twice with PBS. Lastly, fluorescence signal was quantified at Ex485/Em535 nm using a Hidex sense microplate reader (Hidex, Turku, Finland) to detect DCF and normalised against cell number detected from Hoechst dye at Ex355/Em460 nm. Data were normalised against the untreated control.

### 4.8. Calcium Ion Measurement

Intracellular calcium release was measured using a calcium quantification assay kit (Abcam, Cambridge, MA, USA) according to the manufacturer’s instructions. Briefly, HHSECs were first harvested with RIPA buffer to obtain the cell lysates. 50 µL of lysate was mixed with 50 µL of assay reaction buffer containing rhodamine red indicator inside a 96-well black plate and incubated for 15 min in the dark at room temperature. Fluorescence signal was then quantified at Ex535 nm/Em590 nm using Hidex sense microplate reader (Hidex, Turku, Finland) and data was normalised against untreated control.

### 4.9. 3D Co-Culture Model

The 3D co-culture model was constructed using a two-step forced aggregation of LO2 cells and HHSECs seeded separately at 24-h intervals in media. Firstly, LO2 cells were pre-labelled with 5 µM of CellTracker Orange (Life Technologies, Carlsbad, CA, USA) before adding to the non-adherent 96-well plate at a seeding density of twenty thousand cells/well. After 24 h, a spheroid containing LO2 cells was formed. Secondly, the media was carefully removed and replaced with fresh media containing HHSECs pre-labelled with 5 µM of CellTracker Green (Life Technologies, CA, USA) at a seeding density of twenty thousand cells/well for another 24 h. Multiple fluorescence images were taken using Olympus Fluoview FV1000 confocal microscope (Olympus, Tokyo, Japan) and Z-stacking was performed using Imaris software (Bitplane, Belfast, Ireland).

### 4.10. Drug Permeability Study

Sunitinib (LC Laboratories, Woburn, MA, USA) was used as the drug of study due to its intrinsic ability to fluoresce at 540 nm, thus providing an excellent indicator to assess endothelial leakiness [49]. Endothelial cells were first seeded at a density of thirty thousand cells/cm^2^ onto 24-well 6.5 mm transwell insert, polyester filter with 0.4 µm pore (Corning, Corning, NY, USA) for 3 days to form a confluent layer. After 30 min treatment with TiO_2_ NPs, 100 µM of Sunitinib was added into the transwell insert and incubated for another 30 min. Subsequently, 100 µL of media was collected from the lower compartment of the transwell and placed into a 96-well black plate. The fluorescence signal was then quantified using Hidex Sense microplate reader (Hidex, Turku, Finland) at Ex405/Em544 nm. To determine the concentration of transported Sunitinib, a standard curve was plotted with concentrations ranging from 0.625 to 40 µM and compared against the test samples.

### 4.11. Statistical Analysis

Statistical significance was calculated using a 2-tail equal variance Student’s *t*-test comparison, unless otherwise stated. All *p*-values less than 0.05 were considered statistically significant.

## 5. Conclusions

In this study, we showed that TiO_2_ NPs could induce transient endothelial leakiness in liver sinusoids using HHSECs as the endothelial cell model. While TiO_2_ NPs were rapidly internalised into the cells, there were no significant induction of oxidative stress. Instead, we observed a reduction in phosphorylated Akt, which could lead to the detachment of HHSECs. This observation was further supported in our co-culture system, whereby the presence of TiO_2_ NPs resulted in the dispersion of endothelial cells from the hepatocytes. Lastly, we showed that NP-induced leakiness could potentially aid in anti-fibrotic therapy through the enhancement of drug transport across a diseased endothelium. Future studies could be aimed at exploring NP-induced leakiness in other liver diseases and designing novel inorganic NPs to specifically target the endothelial cells lining the liver sinusoids while promoting drug uptake.

## Figures and Tables

**Figure 1 ijms-20-00035-f001:**
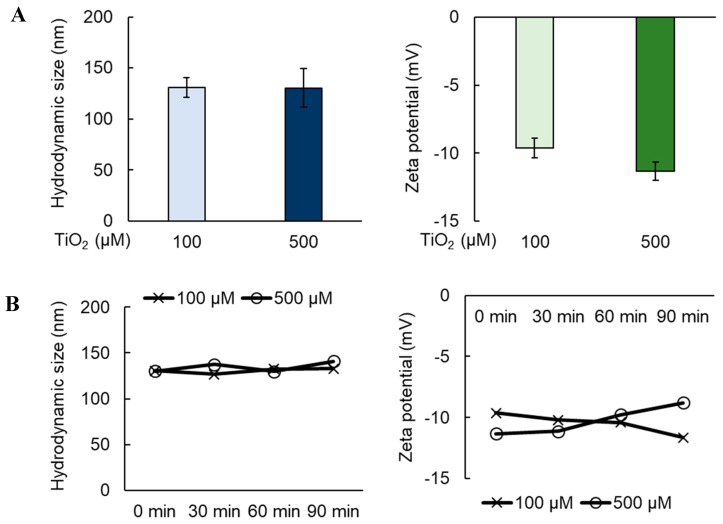
Characterization of titanium dioxide NPs (TiO_2_ NPs) in media. (**A**) Hydrodynamic size (left panel) and surface zeta potential (right panel) of TiO_2_ NPs in media at 0 min. Data represents mean ± SD, *n* = 3. (**B**) The stability of TiO_2_ NPs was measured based on hydrodynamic size (left panel) and surface zeta potential (right panel) up to 90 min post-sonication. Data represents mean ± SD, *n* = 3.

**Figure 2 ijms-20-00035-f002:**
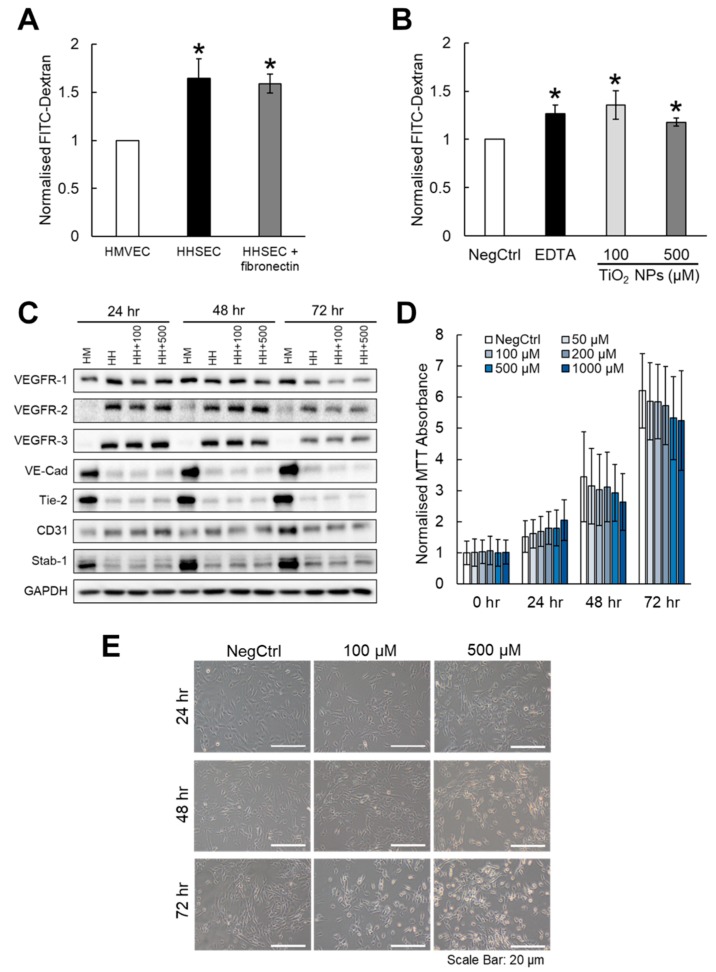
TiO_2_ NPs induced endothelial leakiness in human hepatic sinusoidal endothelial cells (HHSECs) without compromising endothelial biomarkers. (**A**) Transwell permeability assay revealed higher fluorescein isothiocyanate (FITC)-dextran leakiness exhibited by HHSECs compared to human microvascular endothelial cells (HMVECs). Fibronectin coating did not significantly reduce the leakiness of HHSECs. (**B**) TiO_2_ NPs significantly increased the leakiness in HHSECs at two different concentrations of 100 µM and 500 µM, compared to the untreated control (NegCtrl). EDTA was used as a positive control. (**C**) Western blot analyses showed that the exposure of HHSECs to TiO_2_ NPs did not result in observable changes to endothelial biomarkers up to 72 h. Glyceraldehyde 3-phosphate dehydrogenase (GAPDH) was used as a loading control. (**D**) HHSECs treated with various concentrations of TiO_2_ NPs up to 72 h did not exhibit significant difference in cell viability compared to the non-treated control (NegCtrl) at the same time point. (**E**) Treatment of TiO_2_ NPs resulted in observable morphological changes to the cells leading to cellular detachment, particularly after exposure to a higher concentration (500 µM) at the 72-h timepoint. Scale bar = 20 µm. Data represent mean ± SE (*n* = 3), Student’s *t*-test, * *p* < 0.05.

**Figure 3 ijms-20-00035-f003:**
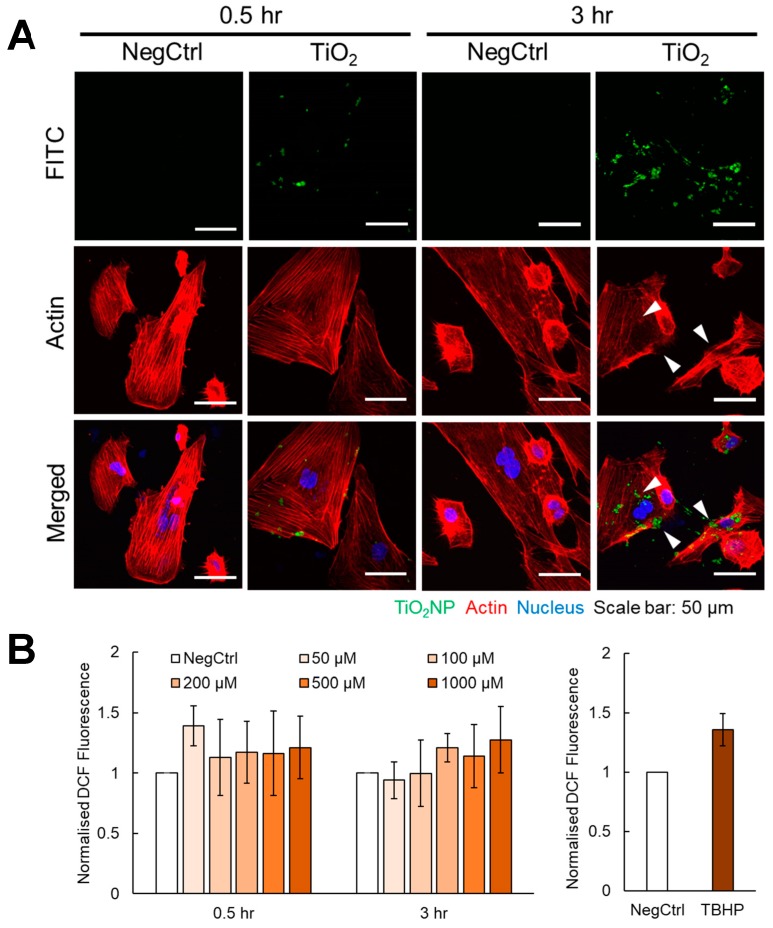
Internalised TiO_2_ NPs did not significantly induce oxidative stress in HHSECs. (**A**) TiO_2_ NPs conjugated with FITC (green) were internalised into HHSECs at two different timepoints of 0.5 and 3 h. White arrows indicate possible disruption of actin fibres. Actin was stained with phalloidin dye (red) and nucleus was stained with Hoechst dye (blue). Scale bar = 50 µm. (**B**) H_2_DCF-DA assay revealed no significant increase in oxidative stress exhibited by HHSECs treated with varying concentrations of TiO_2_ NPs at both timepoints. Tert-Butyl hydroperoxide (TBHP) was used as a positive control. Data represent mean ± SE (*n* = 3).

**Figure 4 ijms-20-00035-f004:**
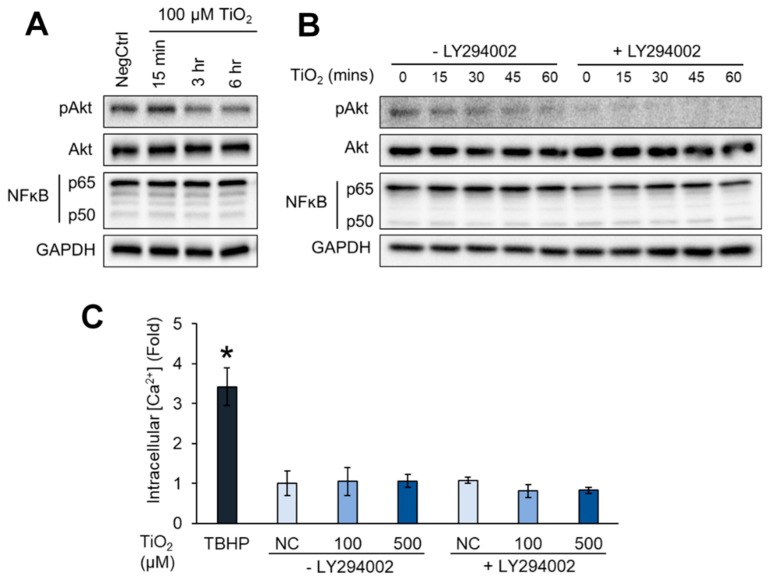
TiO_2_ NPs down-regulated the protein kinase B (Akt) pathway as a late event of endothelial leakiness. (**A**) HHSECs exposed to 100 µM TiO_2_ NPs for 3 and 6 h showed a reduction in phosphorylated Akt. GAPDH was used as a loading control. (**B**) Reduction of phosphorylated Akt showed a time-dependent response upon exposure to 100 µM TiO_2_ NPs up to 1 h. Akt inhibitor (LY294002) was reduced as a positive control to inhibit Akt phosphorylation. (**C**) TiO_2_ NPs did not induce intracellular calcium ion (Ca^2+^) release after 3 h exposure. TBHP was used as a positive control. Data represent mean ± SE (*n* = 3), * *p* < 0.05.

**Figure 5 ijms-20-00035-f005:**
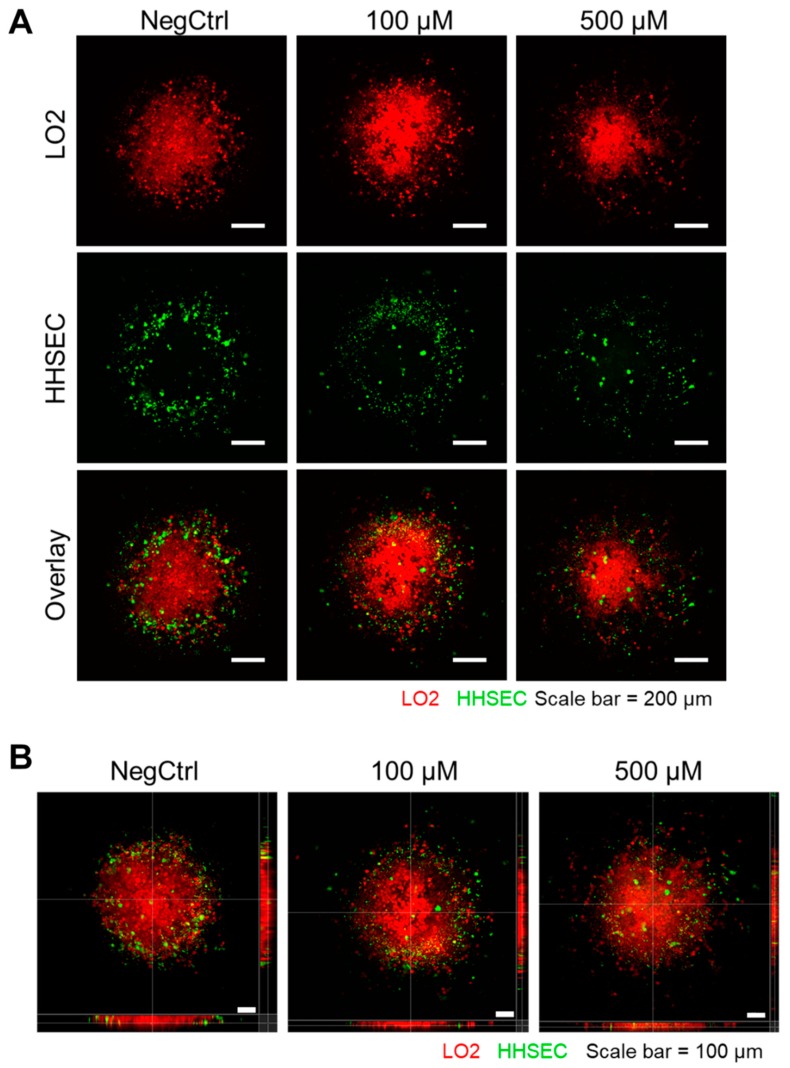
TiO_2_ NPs disrupted the 3D co-culture model of normal human hepatic cells LO2 and HHSECs. (**A**) LO2 cells were stained with CellTracker orange (red) while HHSECs were stained with CellTracker green (Green). TiO_2_ NP treatment resulted in the dispersion of both LO2 and HHSECs away from the core structure. Scale bar = 200 µm. (**B**) Z-stacking of multiple images to reveal a cross-section of the 3D structure showed the absence of HHSECs surrounding the LO2 cells upon exposure to TiO_2_ NPs. Scale bar = 100 µm.

**Figure 6 ijms-20-00035-f006:**
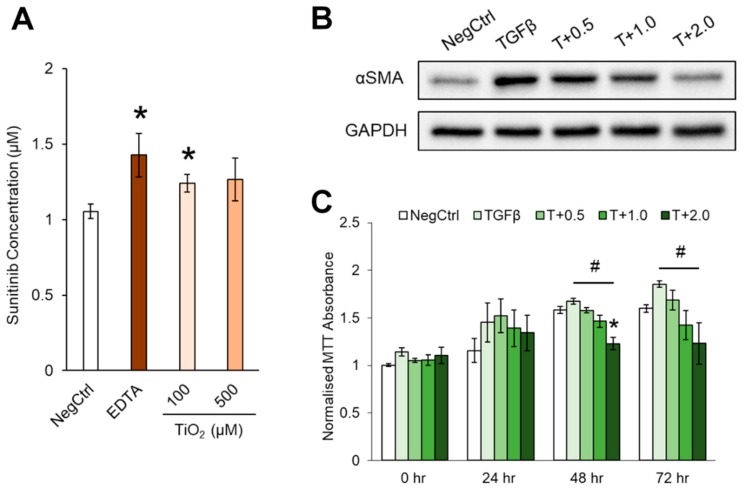
TiO_2_ NP-induced leakiness promoted Sunitinib transport across the HHSEC barrier. (**A**) Endothelial leakiness from HHSECs treated with TiO_2_ NPs significantly increased the transport of Sunitinib across the endothelial barrier, compared to the non-treated control (NegCtrl). EDTA was used as a positive control. Data represent mean ± SE (*n* = 3), one-tail equal variance Student’s *t*-test, * *p* < 0.05. (**B**) The increase in Sunitinib concentration was found to reduce pro-fibrotic marker α-SMA in transforming growth factor-beta 1 (TGF-β1)-activated hepatic stellate cells LX2 after 24 h treatment. Untreated LX2 cells were used as the negative control (NegCtrl). (**C**) Increased concentrations of Sunitinib were also found to reduce the viability of TGF-β1-activated LX2 cells up to 72 h. Data represents mean ± SE (*n* = 3), one-way ANOVA with Tukey HSD test, * *p* < 0.05 denotes significance against non-treated control at respective timepoint, ^#^
*p* < 0.05 denotes significance against TGF-β1 treated control.

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
