# Peer review of "Titanium Dioxide Nanoparticles Enhance Leakiness and Drug Permeability in Primary Human Hepatic Sinusoidal Endothelial Cells"

_ijms, 2018, doi:10.3390/ijms20010035_

Round 1
Reviewer 1 Report
In this study, TiO2 NPs was shown to induce transient endothelial leakiness in liver sinusoids, where HHSECs was used as the endothelial cell model. Also, TiO2 NPs were demonstrated to be internalised into the cells, no inducing oxidative stress was observed. Further development was found to the reduction in phosphorylated Akt of HHSECs. Interestingly, the authors used the co-culture system whereby the presence of TiO2 NPs found in endothelial cells from the hepatocytes. The NP-induced leakiness observation could potentially aid in anti-fibrotic therapy through the enhancement of drug transport across a diseased endothelium.
The work was found to be of great interest for the scientific community. However, there is a minor point which I suggested that the author should change prior publication: Figure 1 should be the characterisation of TiO2 NPs. Otherwise the work should be ready to publish.
Author Response
Thank you for the comments. Previous figure S1 has been shifted to Figure 1 “Characterization of TiO2 NPs in media”. Subsequent figure numbering in the main text and supplementary files have been changed accordingly to the new Figure 1.
Reviewer 2 Report
The article “Titanium dioxide nanoparticles enhance leakiness and drug permeability in primary human hepatic sinusoidal endothelial cells” by Kai Tee et al. is dedicated to an experimental in vitro study of uptake and internalization of titanium oxide nanoparticles by liver cells, and observation of the nanoparticles-induced endothelial leakiness in primary human hepatic sinusoidal endothelial cells. The experimental results presented by authors are important, interesting, and novel. The manuscript is of great quality, with extensive details on experimental methods and techniques used to acquire the data. In my opinion the article should be published in MDPI International Journal of Molecular Sciences, but a certain revision of the article is needed as outlined below:
1. page 3, line 107 section “2.1. TiO2 NPs induced endothelial leakiness in sinusoidal barrier” It is written that “(cells) treated with TiO2 NPs for 30 mins at concentrations of 100 μM and 500 μM”, it is unclear what concentrations are related to – is it molarity of nanoparticle suspension? Based on Titanium oxide concentration? Had the concentration been confirmed by any analytical methods? The authors also omitted this information in page 14 section “4.2. Preparation of TiO2 NPs”. These issues have to be clarified.
2. page 12, line 313 and below. It is written: “TiO2 NPs could have weakened cellular attachment from the substratum. ” The inorganic nanoparticles upon adsorption, are tended to disrupt tertial structure of proteins, especially for electrostatically charged particles like TiO2. The authors are invited to elaborate on this issue.
Author Response
Response 1: Thank you for the comments. Additional details of TiO2 NP preparation “Titanium dioxide nanoparticles (TiO2 NPs) were first prepared by dispersing 5 mg of p25 TiO2 nanopowder (MR: 79.87) with primary size of 21 nm (Sigma-Aldrich, St. Louis, USA) in 1ml of water to give 62.5 mM concentration” has been inserted into the materials and methods”. The stock was further diluted with media into either 100 µM or 500 µM concentration.
Response 2: The experimented TiO2 NPs indeed exhibit a negative zeta potential which could potentially influence the tertiary structure of proteins, especially the surface proteins involved in cell-cell and cell-ECM contacts. However, we postulate that these TiO2 NPs initially reside on top or surrounding the cells after being introduced into the culture system. We also showed that the NPs were rapidly internalized into the cell within 30 mins (Figure 3A), suggesting that any possible effects attributed to endothelial leakiness could be due to the effect of internalized TiO2. While we cannot rule out that the transient influence on cell surface proteins can still trigger cell detachment, there might not be sufficient direct exposure of TiO2 NPs with the substratum for this effect to be complete on its own. Hence, we speculate that there could be additional mechanisms that account for the weakening of cellular attachment (such as interactions with junction proteins). For instance, Tie2 is a receptor tyrosine kinase found on the cell surface which is responsible for mediating cell-cell contacts through the Akt pathway. Hence, Akt pathway is preferentially activated to provide stabilizing signal in endothelial cells. Specifically, in relation to inorganic NPs, SiO2 NPs were previously shown to promote vascular permeability through the inhibition of PI3K/Akt pathway [41]. We also wish to highlight that this effect differs from anoikis in which TiO2 NPs did not result in overt cell death. We postulate that the internalized TiO2 NPs could have played a more major role in perturbing the intracellular Akt pathway (Figure 4A,B) which resulted in the observed event. We have added these points into the discussion section:
“Indeed, inorganic NPs such as SiO2 NPs have been previously shown to disrupt endothelial cell function and promote vascular permeability through the inhibition of phosphoinositide 3-kinase (PI3K)/Akt signalling pathway [41]. Cell-attachment studies on endothelial tyrosine kinase Tie2-mediated anchoring have also reported the significance of Akt pathway to provide a stabilizing signal in the endothelial cells [26], as well as being preferentially activated by dimerized Tie2 in the presence of cell-cell contacts [27]. Furthermore, other studies have reported Akt-dependent mechanisms involving the downregulation of eNOS activity which could direct the loss of cellular attachment from the substratum [42,43]. Taken together, these studies have specifically pinpoint the involvement of Akt in cellular detachment. Therefore, our findings is consistent with these observations whereby a mechano-transduction process may drive TiO2 NP-induced endothelial leakiness towards a structural and a functional change.”